# Efficacy of Health Promotion Interventions Aimed to Improve Health Gains in Middle-Aged Adults—A Systematic Review

**DOI:** 10.3390/geriatrics8030050

**Published:** 2023-04-30

**Authors:** Eunice M. C. P. Santos, Ana M. G. D. S. Canhestro, Jorge M. O. A. Rosário, César J. V. Fonseca, Lara M. G. Pinho, Helena M. S. L. R. Arco

**Affiliations:** 1Institute for Research and Advanced Training, University of Évora, 7000-811 Évora, Portugal; 2Department of Health, Polytechnic Institute of Beja, 7800-295 Beja, Portugal; 3Comprehensive Health Research Centre (CHRC), University of Évora, 7000-811 Évora, Portugal; 4Nursing Department, University of Évora, 7000-811 Évora, Portugal; 5Department of Health Sciences and Technologies, Polytechnic Institute of Portalegre, 7300-555 Portalegre, Portugal

**Keywords:** lifestyle, promoting health, middle-aged adults, health gains, quality of life, well-being

## Abstract

Population aging will be one of the major social transformations in the coming decades, with a very significant impact in all countries. The consequences of this will cause an overload of social and health services. It will be necessary to prepare for an aging population. The promotion of healthy lifestyles is necessary to increase quality of life and well-being as people age. The aim of this study was to identify and synthesize interventions in middle-aged adults that promote healthy lifestyles and translate this knowledge into health gains. We performed a systematic review of the literature with research on the EBSCO Host—Research Databases platform. The methodology followed the PRISMA guidelines, and the protocol was registered with PROSPERO. A total of 10 articles out of 44 retrieved were included in this review, which identified interventions to promote healthy lifestyles with an impact on well-being, quality of life, and adherence to healthy behaviors. The synthesized evidence supports the efficacy of interventions that contributed to positive changes at the biopsychosocial level. Health promotion interventions were educational or motivational and related to physical exercise, healthy eating, and changes in habits and lifestyles related to harmful behaviors (tobacco use, excess carbohydrates in the diet, physical inactivity, and stress). The health gains found were increased mental health knowledge (self-actualization), adherence to physical exercise, improvement in physical condition, adherence to the consumption of fruits and vegetables, increased quality of life, and well-being. Health promotion interventions in middle-aged adults can significantly improve healthy lifestyles, protecting them from the negative effect of aging. For aging to be a successful experience, continuity of healthy lifestyles practiced in middle age is necessary.

## 1. Introduction

Demographic aging is a global reality due to systemically low birth rates and an increased life expectancy. The impact of this aging will be very significant in the coming decades [1]. Aging results in a gradual decrease in physical and mental capacity and an increased risk of disease and death [2]. Active aging is a process of optimizing health opportunities through the participation and security of individuals to increase their quality of life as they age [3]. The importance of active aging is unquestionable, not only to make the last years of life healthier and maintain the functionality of the elderly but also to ensure their active participation in society and a good quality of life. In addition, it must be promoted in a time prior to that in which it is manifested, and investment in this area should be made in an antecedent phase to that which is identified with that of the elderly [3]. The promotion of active and healthy aging is necessary throughout the life cycle [4]. Interventions in this domain and for middle-aged adults need to be effective; therefore, it is important to study how to better promote a healthy lifestyle for successful healthy aging. According to Medical Subject Headings, a middle-aged adult is an adult aged 45–64 years [5]. Preparing for aging is essential to increase quality of life and well-being [6]. Active and healthy aging depends on past health trajectories, as they become determinants of health status and “cognitive dispositions” in relation to health, underlining the importance of promoting healthy behaviors at all ages [7].

Healthy aging encompasses the entire lifespan of a person and requires that a person develop and maintain the functional capacity that allows well-being in old age. Health disadvantages can start early and accumulate throughout life. Intrinsic capacity at any time is determined by several factors, such as underlying physiological and psychological changes, health behavior, and the presence and absence of disease, and is significantly influenced by the environments in which people have lived throughout their lives [8]. Therefore, the adoption of healthy lifestyles is essential to age healthier and with a better quality of life.

Healthy aging can be supported by lifelong education and universal health care that promote safe behaviors and healthy lifestyles throughout life [9]. Health promotion and preventive care, also throughout life, influence the maintenance of maximum functional capacity, which will consequently improve health and well-being [10].

We believe that there are interventions that promote healthy lifestyles that fit into these dimensions and are relevant to the creation of an intervention model for middle-aged adults. According to a study conducted by Canhestro [11], middle-aged adults participate little in interventions to promote healthy lifestyles. That study revealed that personal and sociocultural factors influence the adoption of health-promoting lifestyles and that it is important to reinforce interventions that promote healthy aging throughout life. In particular for middle-aged people, interventions promoting active aging should be combined with behavioral change interventions that favor active participation and adherence of working-age people to interventions. In this context, it is clear that interventions are necessary for the transition to aging to be successful.

The World Health Organization (WHO) [12] advocates a multidimensional model with a positive and biopsychosocial perspective and defines active and healthy aging as a process of optimizing opportunities for health, participation, and security, with the objective of increasing healthy life expectancy and quality of life. The biopsychosocial model of active and healthy aging, i.e., AHA-BPS, based on WHO criteria, allows the monitoring and characterization of the aging of individuals, starting at 50 years of age, evaluating three dimensions, each with two indicators: physical well-being (through frailty and cognition); mental well-being (through self-reported satisfaction with life and non-depressive symptoms); and social well-being (through social participation and social support) [13]. 

However, there are no systematic reviews that responded to this gap. Thus, herein, we conducted a systematic review of the literature to map studies that developed interventions to promote healthy lifestyles among adults, ensuring the rigor, clarity, and quality of the process. The objective of this systematic review was to identify and synthesize interventions for middle-aged adults that promote healthy lifestyles and translate into health gains. This systematic review provides evidence on the nature and effectiveness of interventions to promote healthy lifestyles implemented in middle-aged adults that will bring health benefits in later life, contributing to healthy aging.

## 2. Materials and Methods

### 2.1. Research Strategy

This systematic review was conducted in accordance with the Declaration of Protocols Preferred Reporting Items for Systematic Reviews and Meta-Analyses (PRISMA) [14]. It was registered in the International Prospective Register of Systematic Reviews (PROSPERO) under registration number CRD42022383325.

Considering that the scope of this study is very specific and understudied and because the research question and the objective pertain to intervention effectiveness and health gains, the methodological approach of the included studies was quantitative. In this review, a comprehensive literature search was conducted; the databases consulted were CINAHL Plus with full text, MEDLINE with full text, and Collection of Psychology and Behavioral Sciences.

The search included a combination of several key terms (Medical Subject Headings; (MeSH)): “promoting health”, “health behavior”, “lifestyle”, “outcomes”, “quality of life”, “well-being”, “health related”, middle aged adults”, “middle aged”, “45–64 years”, “randomized control trial”, “randomized controlled trials”, and “rct” using the Boolean operators «AND» and «OR», which forms the Boolean phrase: [(lifestyle OR health behavior OR promoting health) AND (well-being OR quality of life OR outcomes) AND (45–64 years OR middle aged OR middle aged adults) AND (randomized control trial OR randomized controlled trials OR rct)].

The search strategy was adapted to each database and was restricted to the period between 1 January 2017 and 31 August 2022 and publications in English, Portuguese, and Spanish languages.

Initially, an exploratory investigation was performed without limitations. However, given the high number of results, the search was limited to the title, abstract, and/or keywords for each database.

The protocol of this systematic review was registered in December 2022. We started the search in mid-January 2023 and performed the systematic review in February 2023.

### 2.2. Inclusion Criteria

The inclusion criteria for the participants were middle-aged adults who were noninstitutionalized and without cognitive deficits.

We wanted to map out as much information as possible on the health gains obtained through interventions that promote healthy lifestyles, as this information is still scarce in the scientific literature.

For the interventions, studies on any intervention to promote a healthy lifestyle, as well as those to promote well-being, quality of life, and healthy behaviors, in any geographical area and conducted in a community context were included.

This review included primary observational studies and RCTs and was reported according to the Preferred Reporting Items for the Declaration of Protocols for Systematic Reviews and Meta-analyses (PRISMA) [14].

### 2.3. Exclusion Criteria

Studies in which participants were younger than 18 years of age, conducted in hospitals or nursing homes, and published more than five years ago were excluded. 

### 2.4. Quality Assurance

The selection of studies was developed over several phases.

The studies resulting from the search in each database were exported to Mendeley, and duplicates were removed. To minimize bias, two review authors independently assessed the inclusion of studies by reading the title, abstracts, and keywords and excluded studies that did not meet the inclusion criteria for this review. A third review author was consulted in case of disagreement or doubt. Subsequently, the full texts were evaluated. This selection process is illustrated in Figure 1 with the results of the screening in the different phases. Subsequently, we proceeded to the evaluation of the full texts using the same method.

In the data extraction phase, a descriptive evaluation of each study was performed using a not-automatic extraction tool (a table in a Word document) designed to extract information based on the research question (What are the health gains resulting from interventions to promote healthy lifestyles in middle-aged adults?). The following information was collected to address the objective of each study: participants, authors, year, country, methodology, evaluation instruments, intervention, results, and main conclusions.

Data extraction was performed by the same two review authors independently (ES, AC), and any questions or disagreements were resolved by consulting a third review author (HA). 

To assess the quality of the studies and the potential risk of bias, the Joanna Briggs Institute Critical Appraisal Checklist for Randomized Controlled Trials and JBI Critical Appraisal Checklist for Case Series were used [15]. This step was performed by the same two reviewer authors independently, and any disagreement with the quality assessment of the studies was resolved once again by consulting a third review author.

All selected studies met more than 75% of the critical quality assessment criteria of the Joanna Briggs Institute. The results of this evaluation indicate that the selected studies can be used for the synthesis and interpretation of the results [16]. Thus, the quality of the evidence of the studies used in this review is good.

### 2.5. Strategy for Data Synthesis

The synthesis and analysis of the results were narrative in nature and structured to answer the research question. The interventions that were implemented to promote healthy lifestyles in middle-aged adults (45–64 years) and their results are presented by the positive changes at the biopsychosocial level, incorporating the physical, psychological, and social dimensions, according to the active and healthy aging model, i.e., AHA-BPS [13]. This model allows monitoring the evolution of the state of active and healthy aging at the individual level as well as the impact of interventions in favor of active and healthy aging. Active aging has been defined by the World Health Organization as “the process of optimizing opportunities for health, participation, and security, with the aim of improving the quality of life of people as they age” [17]. A table was constructed to summarize the responses to the research question. This representation made it possible to group and synthesize the available data from each study and facilitated the analysis and discussion of the results through the collaboration of all team members.

## 3. Results

The database searches produced 44 results, and after removing duplicate articles and articles outside the period under analysis, 35 publications were identified as eligible. Based on the title and abstract information, 10 articles were selected for a thorough evaluation (Figure 1). The main reasons for exclusion were articles that did not address interventions to promote healthy lifestyles (10 studies) and articles that did not address answers to the research question, i.e., health gains (11 studies).

Ten studies were included in the review [18,19,20,21,22,23,24,25,26,27], four of which were conducted in the USA, two in Taiwan, one in Norway, one in Denmark, one in Iran, and one on the island of Grenada.

Nine randomized control trials [18,19,20,22,23,24,25,26,27] and one case series study [21], which had previously been subjected to a critical evaluation of quality, were selected. The studies included a total of 3230 participants and were conducted in a community setting.

Nine of the included studies evaluated the effectiveness of interventions on the participants’ lifestyles [19,20,21,22,23,24,25,26,27]. Only one of the studies compared the efficacy of various lifestyle interventions [18].

We verified that there is a correlation in terms of the interventions found since the majority focused on healthy eating and physical exercise, and although they were carried out in different continents of the globe, from the USA to Taiwan, passing through Europe, the type of intervention selected and the results obtained were similar. There was also a correlation between the data collection instruments, such as for the assessment of quality of life.

The primary outcomes of these studies were the interventions to promote healthy lifestyles, and the secondary outcomes were the health gains resulting from these interventions.

To facilitate the understanding of each study and to achieve the proposed objectives, as well as answer the research question, we constructed Appendix A (systematic review matrix), which summarizes all the information regarding the included studies, organized by title, origin, year, and authors; participants; method; objective; and interventions used and results.

Interventions to promote healthy lifestyles are essentially educational and motivational interventions of an individual or group nature. Educational interventions are mainly related to physical exercise, healthy eating, and changes in habits and lifestyles related to harmful behaviors (tobacco use, excess carbohydrates in the diet, physical inactivity, and stress). Motivational interventions consisted of motivational interviewing, cognitive behavioral therapy, acceptance and commitment therapy, and personalized communication letters.

The types of interventions extracted from the included studies are shown in Table 1.

Regarding the dimension of the interventions, according to the biopsychosocial perspective, we found that all interventions have an impact at the physical and psychological levels. Only five of the ten interventions have an impact at the social level. 

Table 2 identifies the dimension of interventions from a biopsychosocial perspective.

As for health gains, that is, the way of expressing favorable health outcomes [28], we found that six interventions have an impact on quality of life. In addition to these, there are three more interventions that have an impact on physical condition. Table 3 shows the results of interventions to promote healthy lifestyles, i.e., the health gains.

## 4. Discussion

In fulfillment of the objective mentioned above and with the intention of answering the initial question, this review included studies that allowed a narrative synthesis on the effectiveness of certain interventions that promote healthy lifestyles in middle-aged adults and that translate into health gains.

Most studies focused on the evaluation of the effectiveness of a given intervention on lifestyle and its impact, especially on quality of life.

### 4.1. Interventions That Promote Healthy Lifestyles in Middle-Aged Adults

Unhealthy behaviors are risk factors for noncommunicable diseases such as cardiovascular diseases, cancer, diabetes, and respiratory diseases. Many risk factors for these diseases, such as obesity, tobacco consumption, inadequate diet, stress, and physical inactivity, are modifiable. Interventions to promote healthy lifestyles can be an opportunity to change behavior [29]. Most studies used educational interventions to promote physical exercise [18,19,20,21,22,23,24,25,26,27]. In fact, this intervention type was the one that most contributed to the change in the participants’ lifestyles. There was no significant change in relation to physical exercise in only one study [25]. To promote healthy eating, personalized meal plans were used in seven of the included studies. However, only one of the studies found that such interventions were effective in reducing carbohydrate intake and weight loss [22].

As for the use of technology in interventions, electronic monitoring equipment, such as smartwatches, was utilized in one study [23]. The use of this equipment had a more significant positive effect on lifestyle modifications, verifying an effective strategy in the promotion of public health [23].

All studies used more than one intervention to affect lifestyle, combining educational and motivational interventions [18,19,20,21,22,23,24,25,26,27]. There was no significant difference in the nature of the interventions; however, most interventions were performed in groups [20,21,22,24,25,26].

When comparing the effectiveness of two lifestyle interventions (by sending personalized communication letters or conducting motivational interviews over the phone), personalized communication letters were more effective than motivational interviews by telephone [18]. Regarding motivational interventions, the following were used: motivational interviews [19,21], telephone monitoring [19,22], face-to-face monitoring by professionals such as nurses, physiologists, and coaches [24,25,26], and monitoring by peers [20]. One psychological intervention utilized a smartwatch [23].

These interventions were classified into three dimensions from the biopsychosocial perspective advocated by Bosch-Farré et al. based on the WHO definition of active and healthy aging [13].

### 4.2. Interventions That Translate into Health Gains

With regard to health gains, in the studies selected in this review, the interventions translated into several health gains. In an observational study conducted with adults with, on average, two chronic diseases, a salutogenic approach to behavior change in addition to a motivational interview substantially improved quality of life from baseline after 3 months of follow-up, and improvements were maintained after 15 months of follow-up in all dimensions (except for mental health, for which further improvement occurred) [21]. Another study reported improvements in the quality of life of adults with human immunodeficiency virus (HIV) who participated in a lifestyle modification program for two months. The authors of that study concluded that a lifestyle modification program can be used as an effective intervention to improve the quality of life and immune competence of people with HIV [24].

In a study conducted with adults diagnosed with type 2 diabetes mellitus less than 10 years prior, an intensive lifestyle intervention for one year improved the physical dimension of health-related quality of life. The intervention included counseling from professionals, disease management, increased levels of physical exercise, individualized meal plans prepared by a nutritionist, and supervision by professionals [26]. Another study used a six-month home-based exercise program, supplemented by weekly telephone calls to each participant (female ovarian cancer survivors), conducted by a certified trainer, resulting in improved quality of life in the physical dimension related to health [27]. Based on that study, oncologists and primary care physicians are recommended to refer women with ovarian cancer to physical exercise programs [27].

The duration of an intervention is not always the reason for gains in quality of life. For example, a study conducted with adults, which consisted of an intensive program to promote well-being through the application of cognitive behavioral therapy in addition to acceptance and commitment therapy, lasting two and a half days produced improvements in the participants’ quality of life and well-being over six months [25]. In addition to the above, in another study, intelligent electronic equipment (smartwatches with psychological intervention) was used to shape a positive lifestyle, resulting in improvements in quality of life [23]. Therefore, improved quality of life is an important health gain resulting from lifestyle modification interventions, especially interventions that promote physical exercise, healthy eating, or changes in other harmful behaviors.

Other health gains observed were adherence to healthy behaviors, such as adherence to physical exercise and adherence to a healthy diet, after the implementation of a set of sociocognitive interventions, such as personalized communication letters to promote adherence [18]. There were also gains in grip strength and improvements in physical condition after an intervention implemented for twelve weeks that consisted of a multidisciplinary, individualized aerobic exercise training program, 30 to 50 min per session, combined with motivational telephone interviews once per week [19]. The use of electronic equipment also contributed to other health gains, such as increased participants’ knowledge (self-updating), better stress management, and increased well-being [23].

Health gains resulting from interventions to promote healthy lifestyles were grouped into physical, psychological, and social dimensions based on the AHA-BPS biopsychosocial model of active and healthy aging. The gains in the physical dimension were adherence to physical exercise [18] and adherence to the consumption of fruits and vegetables [18], increased handgrip strength [19], increased cardiorespiratory fitness [19], and weight loss [22]. The gains in the psychological dimension were better stress management [23], increased knowledge [23], and improvements in quality of life [23]. The gains in the social dimension were increases in quality of life [21,23,24,25,26,27] and well-being [25] and an increase in healthy behaviors [20].

When the interventions fit into two dimensions (physical and psychological), gains pertaining to adherence behaviors were highlighted. In addition, health gains were greater and more diversified. When the interventions fit into three dimensions (physical, psychological, and social), the health gains were more comprehensive. Thus, the more complex the intervention, i.e., the more dimensions included in the intervention, the more comprehensive the health gains were. Active training in healthy psychological and behavioral intensification is more effective than the passive provision of guidelines; such interventions should include patient-centeredness, emotional attention, cognitive adjustment, perceived social support, and behavior intensification techniques [30].

The evidence identified in this review allowed the synthesis of a set of interventions for middle-aged adults that promote healthy lifestyles and that translate into health gains. When health promotion initiatives are designed to meet the needs of local communities and priority population groups, such interventions have the potential to positively influence health and well-being within their communities [29].

We performed a comprehensive literature search to systematically search for trials of interventions that promoted healthy lifestyles in middle-aged adults and also intend to summarize the practical implications of these studies. All phases of this review were duplicated, including study selection, risk of bias, and data extraction. Nine of the ten included studies were RCTs that had more than 75% compliance with criteria for assessing quality and potential risk of bias, indicating that they can be used for the synthesis and interpretation of results. Study samples were relatively homogenous.

However, there were limitations in the description of some interventions. For example, the following were not provided in some studies: the cognitive abilities of participants to perform certain tasks, how the instructions to perform the interventions were transmitted, and how the accuracy of participants’ performance of the tasks was guaranteed when the interventions were of an individual nature. Another aspect that was not clear in some studies was the role of the community in interventions, for example, existing resources and partnerships created. Some studies had few participants and were mostly adults of various ages, including middle-aged adults.

The lack of systematic reviews is a limitation of this study; as such, there is no opportunity to compare and discuss the extent to which the results of the current review agree or disagree with those of other reviews. In addition, the search was performed by title, abstract, and/or keywords, and there was difficulty in selecting articles that pertained to the objective of the review, with the languages limited by those familiar to the researchers.

In middle-aged adults, interventions to promote healthy lifestyles designed in projects, plans, or programs addressed the themes of nutrition, physical activity, mental health (management of stress and socializing), and well-being. Gains in health translated into gains in knowledge, improvement in physical condition, improvement in eating habits, and quality of life. Investing in the development and implementation of this type of intervention in middle-aged adults contributes to healthy aging. Projects, plans, and programs to promote healthy lifestyles contribute to training middle-aged adults for active and healthy aging. Table 4 shows the results of the interventions’ themes and the health gains.

Despite the above, it is necessary to continue to make efforts for future revisions in this area to expand the range of effective interventions that contribute to improving the health and quality of life of adults; this is important for the construction of an effective intervention model for adults.

## 5. Conclusions

This systematic review contributes to increasing knowledge on the subject, identifying guidelines for practice based on scientific evidence, and suggesting guidelines for future investigations.

The results of this systematic review serve as a reference for interventions that promote health, more specifically, healthy lifestyles encompassing the three dimensions advocated by the WHO: physical well-being, mental well-being, and social well-being.

Designed to respond to the needs of middle-aged adults to age with better health and quality of life, the synthesized interventions have the potential to influence behavior, lifestyle, knowledge, strength, physical condition, stress, well-being, and quality of life.

Future studies should explore the effectiveness of interventions that promote life satisfaction, participation, and social support, which are also important dimensions that contribute to quality of life as one ages. It would also be interesting to explore interventions that translate into health gains in contrasting populations (different cultures, socioeconomically disadvantaged, and rural and urban populations).

We also suggest, for future research, that literature reviews similar to this one be performed to identify agreements and disagreements.

## Figures and Tables

**Figure 1 geriatrics-08-00050-f001:**
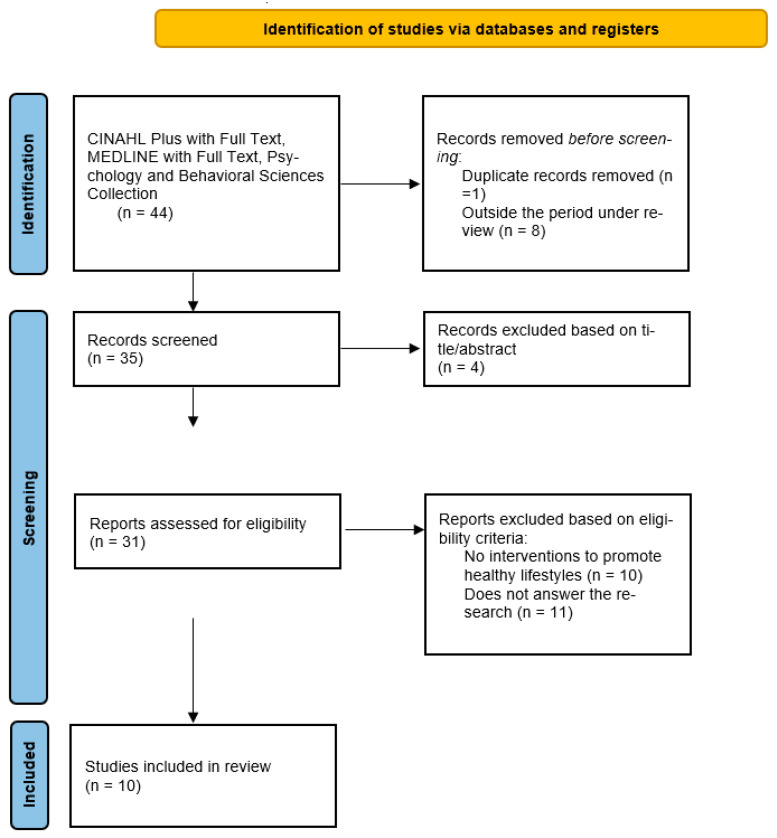
PRISMA Flow Diagram.

**Table 1 geriatrics-08-00050-t001:** Interventions to promote healthy lifestyles that appear in the included studies.

Interventions to Promote Healthy Lifestyles		Studies
Educational	Healthy eating	[18,20,21,22,24,25]
	Physical exercise	[18,19,20,21,22,24,25,26,27]
	Change in habits and lifestyles away from harmful behaviors (smoking, excess carbohydrates, sedentary lifestyle, and stress)	[20,21,22,24]
Motivational	Motivational interviewing	[19,21]
	Cognitive behavioral therapy	[25]
	Acceptance and commitment therapy	[25]
	Personalized communication letters	[18]
Use of electronic equipment	Accelerometers	[21,22]
	Smartwatch/bracelet	[23]
	Polar Watch V800	[24]
Individuals	Performed individually by the person	[18,19,22,23,27]
In group	Performed in interactions with other people in a group	[20,21,22,24,25,26]

**Table 2 geriatrics-08-00050-t002:** Identification of the dimension of interventions to promote healthy lifestyles from a biopsychosocial perspective.

Intervention	Impact of the Intervention	Studies
Intervention over 12 months consisting of the transmission of information about healthy lifestyles (fruit and vegetable consumption and physical exercise) by sending 4 personalized communication letters (TPC)	Physical and psychological	[18]
Intervention implemented across 12 weeks that consisted of a multidisciplinary, individualized aerobic exercise training program, 30 to 50 min per session, combined with motivational telephone interviews, once a week, with duration of 15 to 30 min	Physical and psychological	[19]
Health education conducted by peers over 12 months	Physical, psychological, and social	[20]
Design of a personalized plan for behavior change	Physical, psychological, and social	[21]
Education about healthy lifestyles complemented by telephone calls	Physical and psychological	[22]
Intervention implemented over 3 months that consisted of the use of 2 different commercial smart devices (bracelet and watch)	Physical and psychological	[23]
Promotion of physical and mental exercise and healthy eating (increased consumption of fruit and vegetables) and adoption of healthier habits (physical, mental, and cognitive), over 8 weeks	Physical, psychological, and social	[24]
Implementation of an intensive 2.5-day group program to promote well-being (combining cognitive behavioral theory and acceptance and commitment therapy), followed by periodic follow-up at 6, 12, and 18 months	Physical, psychological, and social	[25]
Implementation of an intensive intervention on lifestyle (physical exercise, nutrition, sleep, management of chronic disease) over 1 year	Physical, psychological, and social	[26]
Program of moderate-intensity aerobic exercises, performed at home, facilitated by weekly telephone calls, over 6 months	Physical and psychological	[27]

**Table 3 geriatrics-08-00050-t003:** Health gains resulting from interventions to promote healthy lifestyles.

Intervention	Health Gains	Studies
Intervention over 12 months consisting of the transmission of information about healthy lifestyles (fruit and vegetable consumption and physical exercise) by sending 4 personalized communication letters (TPC)	Adherence to physical exercise; adherence to the consumption of fruits and vegetables	[18]
Intervention implemented across 12 weeks that consisted of a multidisciplinary, individualized aerobic exercise training program, 30 to 50 min per session, combined with motivational telephone interviews, once a week, with duration of 15 to 30 min	Improvement in physical condition; increased handgrip strength; increased cardiorespiratory fitness	[19]
Health education conducted by peers over 12 months	Increase in healthy behaviors	[20]
Design of a personalized plan for behavior change	Improved health-related quality of life (in all dimensions)	[21]
Education about healthy lifestyles complemented by social telephone calls	Weight decrease	[22]
Intervention implemented over 3 months that consisted of the use of 2 different commercial smart devices (bracelet and watch)	Stress management; increased knowledge (self-actualization); increased quality of life.	[23]
Promotion of physical and mental exercise, healthy eating (increased consumption of fruit and vegetables), and adoption of healthier habits (physical, mental, and cognitive) over 8 weeks	Increased quality of life (in all subscales)	[24]
Implementation of an intensive 2.5-day group program to promote well-being (combining cognitive behavioral theory and acceptance and commitment therapy), followed by periodic follow-ups at 6, 12, and 18 months	Increased quality of life; increased well-being	[25]
Implementation of an intensive intervention on lifestyle (physical exercise, nutrition, sleep, management of chronic disease) over 1 year	Increased health-related quality of life	[26]
Program of moderate-intensity aerobic exercises, performed at home, facilitated by weekly telephone calls, over 6 months	Increased quality of life	[27]

**Table 4 geriatrics-08-00050-t004:** Middle-aged health gains resulting from interventions to promote healthy lifestyles.

Interventions’ Themes	Health Gains
Mental health	Increased knowledge (self-actualization)
Physical activity and physical condition	Adherence to physical exercise
	Improvement in physical condition
Nutrition habits	Adherence to the consumption of fruits and vegetables
Quality of life	Increased quality of life
Well-being	Increased well-being

## Data Availability

Not applicable.

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
