# Peer review of "Efficacy of Health Promotion Interventions Aimed to Improve Health Gains in Middle-Aged Adults—A Systematic Review"

_geriatrics, 2023, doi:10.3390/geriatrics8030050_

Round 1

Reviewer 1 Report

While the authors clearly put a good amount of work into the development of this work, it cannot go without acknowledging that it needs significant revisions before it is ready to be published. 

The keys that need to be improved are:

- Needs strong definition of middle-aged in the paper. Also, within tables and descriptives used throughout, the authors do not state age ranges besides "over 18". There is also no upper limit, such as < 65 yrs old, to describe those that are elderly. 

- Health gains and active aging need clear definitions that are also referenced several times throughout the manuscript to make clear what the authors are talking about. 

- Significant restructuring needs to happen to make sentence structure/overall flow of manuscript easier for the readers. 

Author Response

Dear reviewers

We appreciate all your corrections and suggestions to the proposed research paper: "Interventions to promote healthy lifestyles in middle-aged adults - a systematic review".

We inform you that we have carried out all suggestions.

  • Paragraphs have been improved;
  • Table 1 includes the average age of participants in each study, as we only had this information in one of the studies;
  • We present the definition of Health Gains;

Introduction:

  • We have added the impacts of aging;
  • We explained because the promotion of active and healthy aging is necessary;
  • We have clarified the need for reasons of the search;
  • We fixed the expression "aging as a process";
  • We combined the paragraphs as suggested;
  • Clarified "past trajectories" as health trajectories;
  • At the end of the introduction, the main objective of the article was highlighted (bringing the focus back to interventions and how they impact healthy/active aging);

Materials and methods:

  • We defined "middle-aged adults" (45-64 years);
  • Exclusion criteria presented in a single sentence;
  • We have clarified the extraction tool used;
  • We put the initials of the authors who made each aspect;
  • We explained the value found for assessing the quality of the studies;

Results:

  • We presented the correlation between the studies;
  • Added information about the results;

Discussion:

  • Improved the paragraphs and writing as suggested.

We attach the file with the changes made. All revisions in the manuscript were marked up using the “Track Changes”.

We are available to improve whatever is necessary.

Thank You very much.

With best regards,

Eunice Santos

Ana Canhestro

Jorge Rosário

César Fonseca

Lara Pinho

Helena Arco

Reviewer 2 Report

I have completed my review of the manuscript titled "Interventions to promote healthy lifestyles in middle-aged adults – a systematic review". The work presented in the journal is of high quality and meets the criteria established for the review to be considered adequate. However, I would like the authors to improve a couple of inaccuracies:

- The authors say that one of the inclusion criteria was that the participants were adults. What is the age range that includes the term "adults"?

- The wording of the Discussion needs to be improved. Currently it is a list of ideas/results (very short paragraphs) without a fluid connection between ideas. This is a big weakness.

Author Response

Dear reviewer

We appreciate all your corrections and suggestions to the proposed research paper: "Interventions to promote healthy lifestyles in middle-aged adults - a systematic review".

We inform you that we have carried out all suggestions:

We defined "middle-aged adults" (45-64 years);

Discussion:

  • Improved the paragraphs and writing as suggested.

We have included suggestions from all reviewere in the paper. We attach the file with the changes made. All revisions in the manuscript were marked up using the “Track Changes”.

We are available to improve whatever is necessary.

Thank You very much.

With best regards,

Eunice Santos

Ana Canhestro

Jorge Rosário

César Fonseca

Lara Pinho

Helena Arco

Reviewer 3 Report

The study is very interesting and very attractive in today's society.

A very thorough job has been done and a very exhaustive search of the current literature, but a search in pubmed, wos, scopus is missing, which would be highly recommended.

Author Response

Dear reviewer

We appreciate your suggestions to the proposed research paper: "Interventions to promote healthy lifestyles in middle-aged adults - a systematic review".

We understand the importance of searching on pubmed, wos, scopus. In a future research we will follow this suggestion.

We have included suggestions from all reviewers in the paper. We attach the file with the changes made. All revisions in the manuscript were marked up using the “Track Changes”.

We are available to improve whatever is necessary.

Thank You very much.

With best regards,

Eunice Santos

Ana Canhestro

Jorge Rosário

César Fonseca

Lara Pinho

Helena Arco